# Morphological Characteristics of the Osteoplastic Potential of Synthetic CaSiO_3_/HAp Powder Biocomposite

**DOI:** 10.3390/jfb11040068

**Published:** 2020-09-23

**Authors:** Vladimir Apanasevich, Evgeniy Papynov, Nataliay Plekhova, Sergey Zinoviev, Evgeniy Kotciurbii, Alexandra Stepanyugina, Oksana Korshunova, Igor Afonin, Ivan Evdokimov, Oleg Shichalin, Artem Bardin, Vladimir Nevozhai, Alexandr Polezhaev

**Affiliations:** 1Central Research Laboratory, Institute of Surgery, Pacific State Medical University, 2, Ostryakov Aven., Vladivostok 690990, Russia; oncolog222@gmail.com (V.A.); pl_nat@hotmail.com (N.P.); sinowev@mail.ru (S.Z.); blood416@mail.ru (E.K.); stepanyugina@gmail.com (A.S.); farmaoks@yandex.ru (O.K.); igor23-45@mail.ru (I.A.); bard.pkpb@mail.ru (A.B.); vnevozhay@mail.ru (V.N.); polezhaev@rambler.ru (A.P.); 2Institute of Chemistry, Far Eastern Branch of Russian Academy of Sciences, 159, Prosp. 100-letiya Vladivostoka, Vladivostok 690022, Russia; oleg_shich@mail.ru; 3Far Eastern Federal University, 8, Sukhanova St., Vladivostok 690091, Russia; evdokimov.ivan111@gmail.com

**Keywords:** osteoplastic materials, implant, graft, dental, biocomposite, wollastonite, morphology

## Abstract

The study describes the influence of synthetic CaSiO_3_/HAp powder biocomposite on the process of regeneration in osseous tissue in the alveolar ridges in terms of the morphological characteristics of the osteoplastic potential. The authors investigated the osteoinduction and osteoconduction “in vivo” processes during bone tissue regeneration in the mandible defect area of an experimental animal (rabbit). The possibility of angiogenesis in the graft as an adaptation factor was studied in the process of bone tissue regeneration. The results of the histological study that included the qualitative parameters of bone tissue regeneration, the morphometric parameters (microarchitectonics) of the bone, the parameters of osteosynthesis (thickness of the osteoid plates), and resorption (volume density of the eroded surface) were presented. The results allowed the authors to characterize the possibility of the practical adaptation for synthetic powder biocomposite as an osteoplastic graft for the rehabilitation of osseous defects in dentistry.

## 1. Introduction

Mandibular defects acquired through disease or injury remain among the main and most socially significant problems in dentistry. The solution to this issue requires the reconstruction of the bony skeleton of the face and the long-term rehabilitation of patients [1,2,3]. It includes the prevention of the degradation and atrophy of the alveolar ridge of the mandible due to tooth extraction and the further restoration of teeth with a graft [4].

The restoration of a lost tooth is an integral component in the reconstruction of the dentition that usually involves the application of screw- or cement-retained implants. However, both these methods have certain drawbacks. As a rule, due to the degradation of the bone tissue of alveolar walls [5] and mucositis, there can be a weakening of screw fixations in patients with cement-fixed [6] and screw-fixed constructions [7].

The main problem in the restoration of the dentition during the implantation prosthetic rehabilitation is the atrophy of the lower alveolar ridges [8]. Implantation in the spontaneously healed socket is associated with a high percentage of graft loss [9]. This consistency is also true for implants that were inserted in the formed defect after the removal of a tooth [10]. According to a more effective method, implants should be inserted in a specially prepared area. The socket can be filled with different materials of biological and artificial origin. For example, there are such bone xenografts as cortical-spongioid porcine bone as a powder that contains a different amount of collagen gel (OsteoBiol^®^) [11], decalcified lyophilized bovine bone (Allogro^®^) [12], the analogue of the cellular binding domain of collagen (Peptide-15), mineralized protein-free bone matrix (Bio-Oss^®^), and a mixture of treated cortical and spongioid horse bone graft (Osteoplant^®^) [13,14]. The following artificial materials, such as porous high-density polyethylene (Medpore^®^) [15], three-dimensional bioactive fibreglass carcass with carbonized hydroxyapatite (PerioGlass^®^) [16], pure hydroxyapatite with a structure close to that of bone tissue (Engipore^®^) [17], red algae-derived hydroxyapatite (Algipore^®^) [18], and pure calcium sulfate and phosphate, were proposed [19].

The processes that are observed after transplantation follow a similar algorithm: the graft gets damaged by osteoclasts at a different rate and the stem cells become activated, with their further transformation into osteoblasts [20] that synthesize the bone matrix de novo [21].

Along with the destruction of the graft and osteoinduction, angiogenesis is observed, which plays a key role in the targeted regeneration of a bone [22]. The solution to this problem is as important as the prevention of the epithelial cells migration to the area of regeneration. Angiogenesis proceeds at a high level of expression of VGFR (vascular endothelial growth factor). The effectiveness of controlled regeneration can be evaluated by the density of vessels in the graft [23]. Demineralized lyophilized allografts show a higher rate of resorption. Still, during their induction, the lowest rates of the expression of VGFR and the density of vessels are observed. Calcium hydroxyapatite when added to the autograft bone has been shown to have the maximal parameters of this marker expression.

However, the methods of auto- and allografts reconstruction have the following drawbacks: the need for an additional surgical injury during the graft drawing, even in cases of less invasive methods [24], and the possibility of graft loss [25]. The necessity of the reconstruction of mandible bone tissue that can be critical by volume makes researchers look for a feasible alternative to allografting [26], even though, presently, the application of bone autografts is a preferable method for the regeneration and replacement of lost tissue. Achievements in tissue engineering show brand new possibilities in targeted bone regeneration [27,28]. The implantation of synthetic materials that exert osteoplastic properties in the formed defect of the alveolar ridge is most preferable [29]. The reconstruction of jaws with biomaterials is as effective as bone autografts [30].

Hydroxyapatite (HAp, Ca_10_(PO_4_)_6_(OH)_2_) has a composition and structure similar to that of natural bone [31], has excellent physical and chemical properties (osteoconductivity, bioactivity, resorbability, and slow decay properties), and is considered to be a traditional artificial substitute for bone tissue. One of the most interesting features of HAp is the modification of cell response based on its size: nanometer sizes increase intracellular absorption and reduce cell viability [32]. Additionally, HAp effectively stimulates the production of VGFR endothelial cells identical to neo-angiogenesis in implants. Despite the advantages mentioned, HAp is not considered the best transplant for bone regeneration. Firstly, this is related to the low rate of HAp resorption, which is highly dependent on the degree of dispersion of its particles, negatively affecting the rate of formation of new bone tissue without the loss of the total volume of bone recovered [31]. Secondly, HAp is mechanically unstable in its pure form, which does not allow it to form a strong bone structure, in the volume of which biological fluid should diffuse with the subsequent colonization of mesenchymal stem cells, osteoblasts, and osteoclasts, providing the formation of new bone.

An alternative is promising synthetic materials of composite composition. In particular, the combination of synthetic wollastonite (CaSiO_3_) and HAp is considered to be the most appropriate material in the case. Synthetic wollastonite tends to be osteoconductive, and therefore bioresorption, due to the exchange (removal) of Ca^2+^ and SiO_3_^2−^ ions with the bioorganic environment, is considered as an artificial substitute for bone tissue [33]. Wollastonite has a strong angiogenic potential, which is a crucial factor for the osteointegration of the implant [34,35,36]. Our previously obtained data show the low toxicity and high bioactivity of a similar biocomposite studied in “in vitro” and “in vivo” models [37]. Additionally, in the paper mentioned it was experimentally proved that a biological composite with a macroporous structure has a higher mechanical strength. The rate of its resorption and osseointegration is higher in comparison with that of pure HAp [38], which is similar to the comparison of materials based on tricalcium phosphate (TCP) and their composite forms [39]. Thus, it should be noted that for the targeted tissue engineering of the alveolar bone branches, the mentioned biocomposite may represent a promising perspective, since it is able to provide a long duration of resorption, integration into the bone matrix, the stimulation of the activation of mesenchymal stem cells into osteoblasts, the stimulation of vascular growth, and low toxicity. 

Previous studies conducted by the authors presented original methods for the synthesis of CaSiO_3_/HAp biocomposite as an osteoplastic powder material and ceramic volume matrices with exceptional mechanical characteristics and biocompatible properties [40]. The results of the study on the disperse system obtained by sol-gel (template) synthesis showed that it was not toxic in the “in vitro” model in the conditions for influencing immune cells. Additionally, the authors conducted an experiment to evaluate the osteoinductive potential of CaSiO_3_/HAp powder in the regeneration of bone tissue “in vivo”; it was implanted in an area of an artificially created defect of a laboratory animal (female rabbit) mandible after the removal of an incisor. Preliminary tests showed that the CaSiO_3_/HAp graft actively integrated into the alveolar tissue in the area of the removed tooth; the graft was penetrated by connective tissue and vessels, and the graft also did not cause inflammation or the necrosis of the surrounding bone tissues.

The objective of the study was to analyze in detail the effect of CaSiO_3_/HAp synthetic biocomposite powder on the regeneration of the alveolar jaw ridges of an animal (rabbit) in terms of the morphological characteristics of the osteoplastic potential of the implant. In particular, we explore the identification of possible processes of osteoinduction and osteoconduction, which lead to the recovery of the missing bone tissue in the area of the jaw defect. The results allow us to substantiate the prospects of the practical application of the biocomposite in dentistry as an osteoplastic transplant.

## 2. Materials and Methods 

### 2.1. Synthesis of Biocomposite 

Powder CaSiO_3_/HAp biocomposite was obtained as a structured macropore powder by an original method of sol-gel (template) synthesis proposed earlier by the authors [40]. The method was based on calcium silicate and Hap hydrogel mixture fabrication with a pore-forming template (siloxane-acrylate latex) additive for macroporous structure organization after thermal treatment, according to the scheme; 150 mL of siloxane-acrylate latex water solution (latex: water ratio 1:30) was added to 50 mL of 1.0 M calcium chloride and 50 mL of 1.0 M sodium metasilicate dropwise under intense stirring. Then, the obtained solution was stirred for 3 h at 90 °C until the formation of a dense gel, which was cooled down to room temperature (25 °C) after boiling. After that, 41.5 mL of 1.0 M calcium chloride and 25 mL of 1.0 M ammonia hydrophosphate were added to the obtained gel and stirred for 1 h at room temperature (25 °C).

The synthesis procedure was based on the following chemical reactions:

Wollastonite’s synthesis:

6CaCl_2_ + 6Na_2_SiO_3_ + H_2_O → Ca_6_Si_6_O_17_·2OH + 12NaCl.

Hydroxyapatite synthesis in the calcium silicate solution:

10CaCl_2_ + 6(NH_4_)_2_HPO_4_ + 8NH_4_OH → Ca_10_(PO_4_)_6_(OH)_2_ + 20NH_4_Cl + 6H_2_O.

The obtained gel was filtered, washed with distilled water until the absence of chloride ions was noted, and dried for 5 h at 90 °C until the formation of the amorphous composite xerogel.

At the final stage, the formation of crystalline phases in the material and the removal of the pore-formers from it were carried out by means of thermooxidative treatment in air at 800 °C for 1 h at a heating rate of 5 °C/min in the furnace (Nabertherm GmbH, Lilienthal, Germany).

Ca_6_Si_6_O_17_·2OH/HAp $\overset{800\;{}^{\circ}C}{\to }$ 6CaSiO_3_/Hap + H_2_O ↑.

The characteristics of the studied CaSiO_3_/HAp biocomposite were: powder consisting of medium-size particles (100–250 µm) with a content of HAp of 30 wt%; a bimodal pore structure with meso- and macropores that range from 200 to 500–1000 nm; a specific surface (S_sp._) of 61.7 m^2^/g; does not exert cytotoxicity (the necrosis of neutrophils and macrophages should not exceed 18% and 40%, respectively). 

### 2.2. Biocompatibility Tests

The biocomposite was tested on four 1.5-year-old female rabbits (New Zealand White breed) with a bodyweight of 3 kg. The control group of animals also consisted of four the same female rabbits. The design of the study was planned according to the requirements of the directive of the EU on the protection of animals used for scientific purposes (2010/63/EU). The handling and care of the animals and their withdrawal from the study was performed according to the law on the protection of animals from abusive handling (Chapter V, Article 104679-GD, dated 1 December 1999), and the Declaration of Helsinki (1975 and its 2000 revision). The protocol of the study was approved by the local committee of Pacific Medical State University (protocol No. 1, dated 17 September 2018).

The animals were anaesthetized (Rometar 4.0–6.0 mg/kg, 20 min after Zoletil-50 5–10 mg/kg, i/m) and had an anterior lower left incisor removed. A powder sample with a mass of 0.2–0.25 g with a 0.1–0.5 mm fraction was placed into a formed alveolar socket. After that, the socket was sutured with the non-absorbable monofilm material Prolen (Ethicon, New Brunswick, NJ, USA). To prevent the development of infectious complications, the rabbits received antibiotic therapy with cefotaxime 50 mg/kg (150 mg) i/m BID for 5 days. The animals received anesthesia with tramadol 10 mg/kg for 5 days. The wound surface of the alveolar mucosa was treated with a 0.9% solution of sodium chloride. On day 10, the sutures were removed. Clinical blood analysis and biochemical parameters tests (levels of calcium, phosphorous, magnesium, potassium, sodium, bilirubin, alkaline phosphatase, ALT) were conducted on days 2, 4, 8, 16, and 24. 

The animals were euthanized on day 60 according to the requirements of the Helsinki declaration (i/v injection of potassium chloride under general anesthesia).

### 2.3. Visualization and Morphometric

Biological samples were obtained after removal from the test animals’ bodies by sawing off a fragment of the mandible (3 × 2 cm) containing intact periodontium and part of the alveolar mucosa that was used for the implantation of the biocomposite graft. The sample was fixed in 10% buffered formalin with further decalcification in a decalcifying solution SoftiDec (BioVitrum, Moscow, Russia).

The decalcified biomaterial was used to make waxed blocks and 16–20 µm-thick sections that were stained using Ehrlich’s hematoxylin and eosin with a further histological study with a CX41 microscope, equipped with a digital camera U-TV0.35XC-2 (Olympus, Tokyo, Japan) at a magnification of ×100 and ×400. 

Histological preparations were morphometrically tested according to the requirements of the American Osteological Society [34]. For a complete evaluation of the bone, the authors studied the following parameters: 

1. Parameters that indicated the volume of bone tissue—volume density of trabeculas with the evaluation of the mineralized bone tissue, osteoid, trabecular bone volume (Tb.V, %), and cortical width (Ct.Wi, µm).

2. Morphometric parameters that reflect the microarchitectonics of the spongious bone (trabecular thickness (Tb.Th., µm).

3. Parameter of osteogenesis (osteoid thickness (O.Th., µm), number of osteoblasts (N.Ob., mm^2^).

4. Parameters of resorption—volume density of the eroded surface (ES, %); the number of osteoclasts (N.Oc., mm^2^) calculated per 1 square millimeter of bone section.

Morphometric was performed after 10 measurements for each parameter. For the statistical analysis, we used the Mann–Whitney and Wilcoxon tests. The alveolar holes after the extraction of teeth without grafts were studied as a control test. Eight experimental samples (alveolar holes of lower jaws which included grafts) were analyzed after implantation and extraction.

A morphometric study of the obtained pictures was performed with the Imaging Software CellSens (Olympus, Tokyo, Japan).

The calculation of morphometric parameters was provided using the measuring instruments Zen 2.3 Blue edition (Carl Zeiss GmbH Inc., Oberkochen, Germany). Statistical processing of the experimental results was performed using MS Excel with the subsequent calculation of the arithmetic mean error and Student’s *t*-test. Differences were considered significant at *p* ≤ 0.05.

The ethical approval number for the animal study was No. 3, 20.09.2017 (Pacific State Medical University, Vladivostok, Russia).

## 3. Results and Discussion

The authors studied the tissues that formed the alveolar ridge of the anterior section of the mandible. In the described anatomic area in the mandible of a rabbit, single-rooted teeth (incisors) are located. This anatomic area is shown in multilayer computed tomography (CT) images (Figure 1a–c).

The external and internal surfaces of the compact bones alveolar ridges consist of compact bones and form a cortical plate of the alveolar ridge covered with periosteum. Table 1 shows the results of the morphometric study and the data on the elements of bone tissue from the side of the intact tooth and the graft. 

The cortical thickness on the side of the intact tooth was 231.2 ± 1.8, while on the side of the graft it was 239.9 ± 0.9 (at *p* < 0.05). On the lingual surface, the cortical thickness was larger than on the buccal surface, which agreed with the normal values. In the area of the alveolar ridge, the cortical plate continued in the wall lining the surface of the alveolar compact bone.

In the buccal area of the alveolar arch, the cortical plate of the intact tooth was presented by a compact bone that consisted of osteons; each of them contained from 3 to 10 rows (Figure 2A). The bone surface contained numerous basophil lines of the fusion and foci of periosteocytic resorption (Figure 2B). Uneven thinning of the cortical plate on the major part of the alveolar socket was observed. In places where the minimum bone thickness was observed, the highest volume density of the eroded surface formed with osteoclasts was registered. On the intact side, the volume density of the eroded surface was 11.3 ± 0.6%; on the side of the graft, it was 15.7 ± 1.2% (at *p* < 0.05). The eroded surface was covered with osteoid with no signs of mineralization (Figure 2C). The thickness of the osteoid on the intact side was 2.2 ± 1.6 µm, and on the side of the graft it was 3.1 ± 2.6 µm; the difference was insignificant at *p* ˃ 0.05. In the area of the alveolar fossa, there were lacunas of resorption on the medullar part of the trabecula of the bone (Figure 2D). The medullar part of the bone tissue was filled with loose fibrous connective tissue that contained capillaries. 

After the injection of CaSiO_3_/HAp in the area of the tooth extraction, a complete restoration of the bone outside the alveolar fossa was registered. Morphologically, the area of the exposure of the graft was characterized by the formation of lamellar compact bone that formed the cortical part of the socket (Figure 3A). There were no areas of resorption registered on the medullar part of the trabeculae, unlike the intact tooth. The proliferation of the fibrous tissue spreads on the medullar part of the bone. There were single foci of periosteocytic resorption observed. The mean thickness of the cortical plate was uniform and equal to 293.9 ± 0.9 µm. In the periosteum, there were foci of neoangiogenesis registered that were presented as full-blooded capillaries. There were vast areas of reactive fibrosis, which was visually characterized by an increase in the mean thickness of the periodontal ligament (Figure 3B). 

In the compact bone that surrounds the alveolar socket after the filling with powder graft, the number of osteons increased close to the alveolar lumen, as well as on the intact side (Figure 3C). The periodontal ligament was the source of the newly generated connective tissue in the graft because it was rich in the stem cells that transformed into the endotheliocytes and pericytes of the newly formed vessels [31]. Further, the pericytes in bone tissue are capable of transformation into osteoblasts [37], and this fact can be interpreted as an osteoinductive phenomenon for a disperse CaSiO_3_/HAp biocomposite as a structural powder. Ions of calcium and phosphate are osteoinductive factors for mesenchymal cells that stimulate their transformation into osteoblasts and contribute to the genesis of new bone tissue [38]. The graft acted as a component of a regenerative process of bone tissue. Granulating tissue penetrated the material of the graft, organized it, and fixed it to the periodontium (Figure 3D). This peculiarity of CaSiO_3_/HAp was considered to be an osteoconductive factor. Along with this, the majority of the authors believe that it is necessary to prevent the development of fibrous tissue in the area of the graft.

The periosteum of the external and internal surface of the alveolar ridge was presented as a thin layer with poor vascularization. It differs from the alveolar periodontal ligament that contacts the alveolar periosteum. At the same time, the material that lines the alveoli from the inside contains an increased number of full-blooded vessels. The regional hyperemia of the spongious bone adjacent to the alveoli is observed. An increased number of full-blooded vessels (Figure 3B) represents it. This indicates an important role of angiogenesis in the regulation of structural homeostasis in this region in the mandible. At the same time, the blood vessels of this section of the alveolus have a wide lumen and a thin wall with no adventitia. In the graft samples, there was a tendency toward a thinning of minor parts of areas of cortical plate in the compact bone that lines the alveolar socket (from 142.3 to 274.1 µm). The appearance of the massive trabeculae of the spongious bone adjoining the cortical plate in the area of the alveolar socket was observed. The thickness of the trabeculae of the spongious bone was 52.1 ± 3.9 µm, and in the area of implantation it was 68.5 ± 6.5 µm (*p* < 0.05) (Figure 3A).

The morphometric study of the components of bone tissue in the area of implantation and the control samples showed that the number of osteoblasts in the area of implantation per 1 mm^2^ was significantly higher than in the control area, and was equal to 1.6 ± 0.12 vs. 1.1 ± 0.1. The number of osteoclasts in the area of implantation was higher and was equal to 0.4 ± 0.28 vs. 0.1 ± 0.05 in the control sample per 1 mm^2^, but this difference was statistically insignificant. These data agree with the observation [41] wherein the wollastonite fibres in the composition of cement induced the differentiation of osteoblast-like cells. A vast review showed that the presence of silica stimulated the development of the bone matrix, collagen type I, and the differentiation of osteoblasts, and contributed to the settlement of hydroxyapatite. The mechanism and characteristics of the interaction of structural component, and the phase of bone tissue neoplasm using silicates are given in the review [42,43]. At the same time, the presence of silicates in the bone tissue suppressed the activity of osteoclasts [44,45], which agreed with the authors’ observations in the present study. 

The biochemical parameters (the levels of phosphorus, magnesium, potassium, sodium, bilirubin, alkaline phosphatase) did not differ from the normal values typical for an adult animal included in the experiment. The level of calcium was 4.65 ± 0.44 mmol/L, which was higher than normal (2.4–4.2 mmol/L). At the same time, before the experiment the level of calcium was within the normal range (4.09 mmol/L). In addition, the level of ALT increased in comparison with the normal values (417–1000 nkat/L) to 2012 ± 111 on day 8, with a tendency towards a decrease by day 24 of the experiment (1010 ± 124 nkat/L). The mean values of ALT were 1319.6 ± 218.5 nkat/L, which were higher than the normal values. 

After the implantation into the alveolar socket of the removed tooth, the formation of dense amorphous material was observed in the alveolar lumen, with signs of ossification. Unlike dentine and pre-dentine, the substance that formed the graft did not contain osteons. It was evident that the metabolism of the formed bone matrix occurred due to the vessels of the fibroreticular tissue and adjoining osteocytes, which prevented the sequestration of the graft. 

After the implantation, an expressed regeneration of alveolar tissues was observed. Loose fibrous connective tissue penetrated the graft substance. It contained blood vessels, cells, fibrillary material, and an amorphous component of the connective tissue. The substance of the graft was densely aggregated in the structural complex that did not contain fibrillary material. The regenerate contained blood vessels of the sinusoid type.

Thus, during the implantation of the removed tooth in the alveolar socket, the events of the protective reactions of regeneration are observed with no signs of osteitis and periostitis in the mandible. The growth of vessels and connective tissue in the graft occurs without the thinning and resorption of the bone trabecular in the region adjoining the incisor alveoli. Osteoporotic shifts in the studied bones were not revealed. The lack of condensing osteitis was proved by the lack of bone compaction (osteosclerosis), which preserved the spongious structure and had well-visualized osteons and intramedullary spaces. The lack of fibrous osteitis or fibrous osteodystrophy indicated that there was no progressing decalcification of bones with the further replacement of bone tissue elements from fibrous tissue. This indicated the positive dynamics of the implantation in the alveolar socket [46].

## 4. Conclusions

In “in vivo” studies, it has been found that a synthetic CaSiO_3_/HAp transplant activates osteoinduction and osteoconduction, which leads to the recovery of missing bone tissue in the area of jaw defects in laboratory animals. Angiogenesis, which is an adaptation factor in bone regeneration, has been identified in the transplant body. The osteoplastic biocomposite studied can be recommended for bone defect recovery in dentistry. For instance, it can be used to prevent the thinning of the paradontium and the cortical plate of compact bone and to avoid a loss of bone tissue in the defect area. The deformation of the jaw dental arch and bite changes can be minimized.

## Figures and Tables

**Figure 1 jfb-11-00068-f001:**
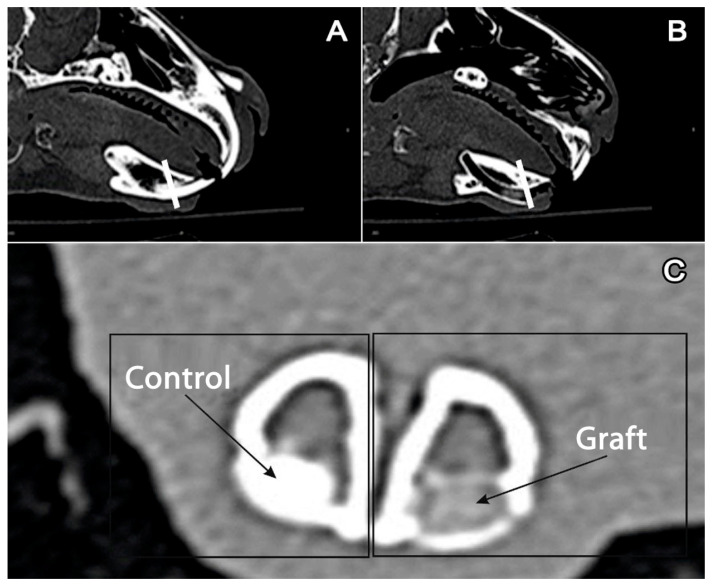
Multilayer computed tomography of the mandible: (**A**) intact side, a white line goes through the incisor. (**B**) The side with a removed incisor and graft in the alveolar fossa, a white line goes through the graft. (**C**) Cross-section of the mandible. Left: preserved incisor (control); right (graft): graft in the alveolar fossa.

**Figure 2 jfb-11-00068-f002:**
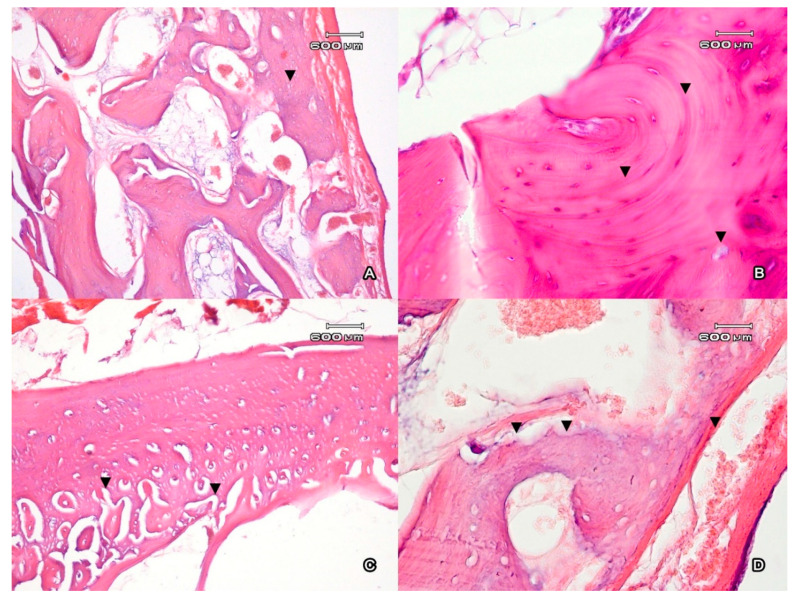
The cortical plate (**A**) of the incisor alveolus represented by the compact bone (▼). Magnification ×100. The compact bone (**B**) contains signs of active transformation (basophil lines of the (▼)). Periosteocytic resorption (▼). Magnification ×400. The external surface of the alveolar edge (**C**) of the mandible next to the area of implantation of CaSiO3/HAp. The number of osteons increased close to the alveolar lumen in the compact bone (▼). Magnification ×100. (**D**) The stripe of osteoid on the bone surface facing the alveolar fossa (▼). Lacunas of absorption on the medullar part of the trabecula of the bone (▼). Magnification ×400. Ehrlich’s hematoxylin and eosin stain.

**Figure 3 jfb-11-00068-f003:**
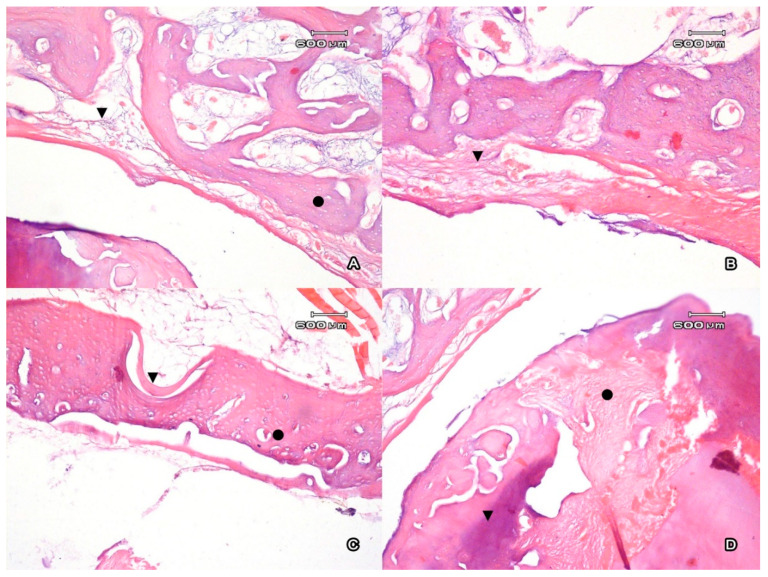
(**A**) The cortical cross-section plate (●) of the walls of the alveolar fossa of the incisors after CaSiO3/HAp implantation. The proliferation of fibroreticular tissue with the spread on the medullar part of the bone (▼). Magnification ×100. (**B**) Internal surface of the alveolar edge of the mandible next to the area of the CaSiO3/HAp implantation. An increase in the thickness of the periodontal ligament (▼). Magnification ×100. (**C**) Periosteum of the internal surface of the alveolar fossa, the filling of the osseous defect with osteoid (•).The highest number of osteons was observed close to the internal surface of the alveolar fossa (▼). Magnification ×100. (**D**) The ossification of the implant in the lumen of the alveolar fossa (•). Filling of the cavities with loose fibrous connective tissue with angiomatosis (▼). Magnification ×100. Ehrlich’s hematoxylin and eosin stain.

**Table 1 jfb-11-00068-t001:** Morphometric characteristics of the bone tissue.

Parameters	Alveolar Wall without Graft *n* = 8	Alveolar Wall with Graft *n* = 8
CW (the width of the cortical layer width, µm)	231.2 ± 1.8 ^1^	293.9 ± 0.9 ^1^
TT (trabeculae thickness, µm)	52.1 ± 3.9 ^1^	68.5 ± 6.5 ^1^
N.Ob (quantity of osteoblasts, mm^2^)	1.1 ± 0.1 ^1^	1.6 ± 0.12 ^1^
OTh (osteoid thickness, µm)	2.2 ± 1.6 ^2^	3.1 ± 2.6 ^2^
ES (surface erosion, %)	11.3 ± 0.6 ^1^	15.7 ± 1.2 ^1^
N.Oc (quantity of osteoclasts, mm^2^)	0.1 ± 0.05 ^2^	0.4 ± 0.28 ^2^
TbV (mineralized volume, %)	53.5 ± 3.8 ^1^	49.2 ± 4.3 ^1^

^1^ if *p* ≤ 0.05, the differences in relation to the control group are reliable. ^2^ if *p* ≥ 0.5, the differences were considered statistically unreliable.

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
