# Peer review of "Morphological Characteristics of the Osteoplastic Potential of Synthetic CaSiO3/HAp Powder Biocomposite"

_jfb, 2020, doi:10.3390/jfb11040068_

Round 1

Reviewer 1 Report

This manuscript presents the morphological characteristics of the osteoplastic potential of synthetic CaSiO3/HAp powder biocomposite. In general, this paper is interesting and well-organized. The authors should consider the following comments:

Comments 1: All references need to be carefully checked and revised. In the reference section, authors should pay attention to formulate a unified format of article titles, doi and page numbers.

Comments 2: Please add the scale bar in Figure 2 and Figure 3.

Comments 3: The assessment of cell cytocompatibility did not only include the cell viability, but also include the whole cell shape. The authors should be recommended to provide SEM images of cells shape.

Comments 5: In Introduction section, authors should provide more detail why they choose HAp to prepare biocomposite. Some literature may be helpful: https://doi.org/10.1016/j.msec.2015.11.040; https://doi.org/10.1002/adfm.201903055

Comments 4: The authors mentioned “Granulating tissue penetrated the material of the graft, organized it, and fixed it to the periodontium (Figure 3D)”. However, Figure 3D was not found in this manuscript.

Comments 5: What are the statistical tests that are being used for the statistical analysis?

Author Response

Dear Reviewer,

We deeply appreciate the time you spent reviewing our paper and valuable recommendations you made. All the comments are taken into account and corresponding changes are made to the manuscripts body text. Detailed point-by-point answers are presented below.

Reviewer #1.

This manuscript presents the morphological characteristics of the osteoplastic potential of synthetic CaSiO3/HAp powder biocomposite. In general, this paper is interesting and well-organized. The authors should consider the following comments:

Comment #1.  All references need to be carefully checked and revised. In the reference section, authors should pay attention to formulate a unified format of article titles, doi and page numbers.

Response to comment #1:

Thank you for the comment. All changes for references have been provided (Line 349-490).

Comment #2. Please add the scale bar in Figure 2 and Figure 3.

Response to comment #2:

All figures have been corrected.

Comment #3. The assessment of cell cytocompatibility did not only include the cell viability, but also include the whole cell shape. The authors should be recommended to provide SEM images of cells shape.

Response to comment #3:

Thank you very much for your valuable comment! We strongly agree that microscopic addition is an important and useful for this investigation. However, for this study, at this stage, it is impossible to make an SEM, since the accumulated bioassays were all consumed after all analysis. We ask the distinguished reviewer to treat this with understanding and to accept this manuscript without these results. At the same time, this experiment continues and the biocomposite is studied during of implantation of a large experimental animal (pig), which will allow us to conduct SEM studies and present them as further result.

Thanks in advance for understanding!

Comment #4. The authors mentioned “Granulating tissue penetrated the material of the graft, organized it, and fixed it to the periodontium (Figure 3D)”. However, Figure 3D was not found in this manuscript.

Response to comment #4:

Thanks for the comment. All figures have been corrected.

Comment #5. In Introduction section, authors should provide more detail why they choose HAp to prepare biocomposite. Some literature may be helpful: https://doi.org/10.1016/j.msec.2015.11.040; https://doi.org/10.1002/adfm.201903055

Response to comment #5:

The “Introduction” has been expanded and the necessary explanation and references have been added (Line 79-106).

The references have been included [30] (Line 88) and [38] (Line 102).

Comment #6. What are the statistical tests that are being used for the statistical analysis?

Response to comment #6:

Morphometric was performed after 10 measurements for each parameter. The calculation of morphometric parameters was provided using measuring instruments Zen 2.3 (Blue edition) “Carl Zeiss GmbH inc.” (Germany). Statistical processing of the experimental results was performed using MS Excel with subsequent calculation of the arithmetic mean error and Student's t-test. Differences were considered significant at p ≤ 0.05.

The addition information has been added (Line 194-197).

Reviewer 2 Report

This manuscript is a study of the osteoplastic potential of a synthetic CaSiO3/HAp powder biocomposite. The topic is highly relevant and knowledge is needed. However, there are some issues that need to be addressed.

Line 20  - “The authors evaluated the activation possibilities of the osteoinduction process and osteoconduction during the restoration of the volume in the bone tissue in the area of the mandible defect of a laboratory animal” please rephrase this sentence.

Line 43 – “the lower alveolar ridges” do you mean mandibular ridges?

Line 62 – “The solution to this problem”  - what problem do you mean?

Line 86 “Synthetic wollastonite is prone to osteoconductivity and bioresorption due to the exchange of 86 bioorganic medium with Ca2+ and SiO32-, because of this it is considered as an artificial graft for bone tissue.” – this sentence is unclear.

Line 88 – “Moreover, wollastonite exerts an expressed angiogenic potential, which is the key position for the osteointegration of the graft” – pleas add a bibliography to support this thesis.

Line 90 – “This occurs because hydroxyapatite is characterized by expressed osteoinductive and osteoconductive properties.” - – pleas add a bibliography to support this thesis.

Line 91 – “It stimulates the appearance of VGFR expressing endothelial cells, which is identical to neoangiogenesis in a graft.” – what do you mean by “appearance”?

Line 104 - CaSiO3/Hap -  please change into - CaSiO3/Hap

Line 130 – “Powder graft (0.2-0.25 g) with 0.1-0.5 mm fraction” – the sentence is unclear

Line 137 and 139 – Days – please change into days

Line 155 - Ct.Wi, μm). – please add “(“

Lines 153-160 – please use one consisted way of using the brackets

Line 170 – “alveolar well” do you mean alveolar wall?

Lines 172 – “2 statistically significant differences in relation to the control group when Ñ€≥0,5” - what kind of statistical analysis was used and how is it possible that the statistical significance was noted for Ñ€≥0,5? What was the control group? How did you manage to analyze the statistical significance of a group of 4 rabbits?

Fig. 2. A. Magnification is missing. What do the arrows point at?

Fig.2. C. It is difficult to understand which arrows point at which feature.

Fig.2.D. It is difficult to understand which arrows point at which feature.

Line 210 – “Morphologically, the area of the exposure of the graft was characterized by the formation of lamellar compact bone that formed the cortical part of the socket (Figure 3A)” – in Fig.3 legend, Fig 3A is reported as the intact side and 3B as a side with the graft.

Fig.3. – “C” is missing on the third figure.

Line 217 – it is challenging to see the increased thickness of the periodontal ligament on Fig.3B same as  the number of osteons on Fig. 3C – Line 225. It seems that Figure 3 is not the figure that is described in the text

Line 234 – there is no Fig. 3D. It seems that Figure 3 is not the figure that is described in the text.

Line 241 and 249 – The figure that is mentioned in the text seems to be missing.

Lines 252 and 253 – the “±” is converted in the pdf into some other symbol

Line 256 – “A vast review showed that the presence of silica stimulated the development of bone matrix, collagen type I, differentiation of osteoblasts, and contributed to the settlement of hydroxyapatite.” - pleas add a bibliography to support this thesis.

It would be beneficial to change “implant” in the text into “graft”

Line 282 – “absorption” should it be “resorption”?

Line 296 – “Thus, it was shown that the studied CaSiO3/HAp biocomposite could be recommended for the application as an osteoplastic material for the rehabilitation of bone defects,  for example, in dentistry, for the prevention of thinning of the periodontium and cortical plate of the compact bone, as well as a decrease in the volume of the bone tissue in the area of the defect, which,  in turn, leads to the deformation of dental arches and alteration of occlusion.” It would be beneficial to divide this sentence into 2 or 3 shorter sentences.

Author Response

Response to the Reviewers’ Comments

Dear Reviewer,

We deeply appreciate the time you spent on reviewing our paper and valuable recommendations you made. All the comments are taken into account and corresponding changes are made to the manuscripts body text. Detailed point-by-point answers are presented below.

Reviewer #2.

This manuscript is a study of the osteoplastic potential of a synthetic CaSiO3/HAp powder biocomposite. The topic is highly relevant and knowledge is needed. However, there are some issues that need to be addressed.

Comment #1. Line 20  - “The authors evaluated the activation possibilities of the osteoinduction process and osteoconduction during the restoration of the volume in the bone tissue in the area of the mandible defect of a laboratory animal” please rephrase this sentence.

Response to comment #1:

Correction has been provided (Line 20-22):

“The authors investigated of the osteoinduction and osteoconduction “in vivo” processes during of the bone tissue regeneration in the mandible defect area of an experimental animal (rabbit)”.

Comment #2. Line 43 – “the lower alveolar ridges” do you mean mandibular ridges?

Response to comment #2:

Should be used considered “the alveolar ridge of lower jaw” (Line 43).

Comment #3. Line 62 – “The solution to this problem”  - what problem do you mean?

Response to comment #3:

Thank you very much for the comment. This sentence is indicated in this context by mistake. The sentence has been deleted.

Comment #4. Line 86 “Synthetic wollastonite is prone to osteoconductivity and bioresorption due to the exchange of bioorganic medium with Ca2+ and SiO32-, because of this it is considered as an artificial graft for bone tissue.” – this sentence is unclear.

Response to comment #4:

The correction of sentenced and necessary references have been provided (Line 94-96):

“Synthetic wollastonite is prone to osteoconductivity and bioresorption due to the exchange of bioorganic medium with Ca2+ and SiO32-. During this biochemical process the bioactive apatite and its derivatives can be produced on the wollastonite surface [32]. Due to this such the synthetic implant is prospective  for the wide applicant as an artificial graft for bone tissue”

Comment #5. Line 88 – “Moreover, wollastonite exerts an expressed angiogenic potential, which is the key position for the osteointegration of the graft” – pleas add a bibliography to support this thesis.

Response to comment #5:

The additional references [doi:10.1016/j.tox.2011.06.016.; https://doi.org/10.1155/2019/1202159; doi: 10.1007/s10561-011-9285-x] have been added in the manuscript [33] [34] [35] (Line 97).

Comment #6. Line 90 – “This occurs because hydroxyapatite is characterized by expressed osteoinductive and osteoconductive properties.” - – pleas add a bibliography to support this thesis.

Response to comment #6:

[https://doi.org/10.1155/2019/1202159; doi: 10.1007/s10561-011-9285-x] have been added in the manuscript [34] [35] (Line 97).

Comment #7. Line 91 – “It stimulates the appearance of VGFR expressing endothelial cells, which is identical to neoangiogenesis in a graft.” – what do you mean by “appearance”?

Response to comment #7:

It means “producing of VGFR…”. The correction has been done (Line 84).

Comment #8.  Line 104 - CaSiO3/Hap -  please change into - CaSiO3/Hap

Response to comment #8:

Correction has been done.

Comment #9. Line 130 – “Powder graft (0.2-0.25 g) with 0.1-0.5 mm fraction” – the sentence is unclear

Response to comment #9:

Correction has been done (Line 162):

“Powder sample with mass 0.2-0.25 g with 0.1-0.5 mm fraction was placed into a formed alveolar socket.”

Comment #10. Line 137 and 139 – Days – please change into days

Response to comment #10:

Correction has been done (Lines 169, 171, 310).

Comment #11. Line 155 - Ct.Wi, μm). – please add “(“

Response to comment #11:

Correction has been done (Line 187).

Comment #12. Lines 153-160 – please use one consisted way of using the brackets

Response to comment #12:

Correction has been done (Line 186-193).

Comment #13. Line 170 – “alveolar well” do you mean alveolar wall?

Response to comment #13:

Absolutely right - Correction has been provided (Line 222).

Comment #14. Lines 172 – “2 statistically significant differences in relation to the control group when Ñ€≥0,5” - what kind of statistical analysis was used and how is it possible that the statistical significance was noted for Ñ€≥0,5? What was the control group? How did you manage to analyze the statistical significance of a group of 4 rabbits?

Response to comment #14:

Morphometric was performed after 10 measurements for each parameter. For statistical analysis we used the Mann-Whitney and Wilcoxon tests. The alveolar holes after extraction of teeth without graft were studied as a control test. For 8 experimental samples (alveolar holes of lower jaws which included graft) after implantation and extraction were analyzed.

If p≥0.5, the differences were considered statistically unreliable.

Correction has been done (Line 194-197 and 223-224).

Comment #15. Fig. 2. A. Magnification is missing. What do the arrows point at?

Response to comment #15:

All corrections have been done (Line 228-236).

Comment #16. Fig.2. C. It is difficult to understand which arrows point at which feature.

Response to comment #16:

All corrections have been done (Line 228-236).

Comment #17. Fig.2.D. It is difficult to understand which arrows point at which feature.

Response to comment #17:

All corrections have been done (Line 228-236).

Comment #18. Line 210 – “Morphologically, the area of the exposure of the graft was characterized by the formation of lamellar compact bone that formed the cortical part of the socket (Figure 3A)” – in Fig.3 legend, Fig 3A is reported as the intact side and 3B as a side with the graft.

Response to comment #18:

All corrections have been done (Line 248-257).

Comment #19. Fig.3. – “C” is missing on the third figure.

Response to comment #19:

All corrections have been done (Line 248-257).

Comment #20. Line 217 – it is challenging to see the increased thickness of the periodontal ligament on Fig.3B same as  the number of osteons on Fig. 3C – Line 225. It seems that Figure 3 is not the figure that is described in the text

Response to comment #20:

All corrections have been done (Line 248-257).

Comment #21. Line 234 – there is no Fig. 3D. It seems that Figure 3 is not the figure that is described in the text.

Response to comment #21:

All corrections have been done (Line 248-257).

Comment #22. Line 241 and 249 – The figure that is mentioned in the text seems to be missing.

Response to comment #22:

All corrections have been done (Line 248-257).

Comment #23. Lines 252 and 253 – the “±” is converted in the pdf into some other symbol

Response to comment #23:

Correction has been done (Line 295-296).

Comment #24. Line 256 – “A vast review showed that the presence of silica stimulated the development of bone matrix, collagen type I, differentiation of osteoblasts, and contributed to the settlement of hydroxyapatite.” - pleas add a bibliography to support this thesis.

Response to comment #24:

The necessary information has been added in the text (Line 301-303):

“The mechanism, characteristics of the interaction of structural components and the phase of bone tissue neoplasm using of silicates are given in the review [doi: 10.1155/2013/141427].

Comment #25. It would be beneficial to change “implant” in the text into “graft”

Response to comment #25:

Correction has been done.

Comment #26. Line 282 – “absorption” should it be “resorption”?

Response to comment #26:

Absolutely right - Correction has been provided (Line 326).

Comment #27. Line 296 – “Thus, it was shown that the studied CaSiO3/HAp biocomposite could be recommended for the application as an osteoplastic material for the rehabilitation of bone defects,  for example, in dentistry, for the prevention of thinning of the periodontium and cortical plate of the compact bone, as well as a decrease in the volume of the bone tissue in the area of the defect, which,  in turn, leads to the deformation of dental arches and alteration of occlusion.” It would be beneficial to divide this sentence into 2 or 3 shorter sentences.

Response to comment #27:

Thank you for the comment. The sentence has been divided (Line 334-340).

Reviewer #3.

The manuscript evaluated the influence of synthetic CaSiO3/HAp powder biocomposite on the process of regeneration in the osseous tissue in the alveolar ridges in terms of the morphological characteristics of osteoplastic potential.

The study is of sound design and of clear practical and clinical interest, but some improvements are needed. I suggest accepting this article with minor revision.

I would give some  comments and suggestions, which will help to improve the quality of this manuscript.

Comment #1. Introduction. please describe the difference and novelty compared to the previous studies(other authors)

Response to comment #1:

The necessary information has been added in the manuscript “Introduction” (Line 79-106).

Comment #2. Introduction. please clarify and emphasize the objective of this study (last two sentences of the introduction section)

Response to comment #2:

The necessary correction has been added in the “Introduction” (Line 118-123).

Comment #3. Introduction. page 2 74-76. I recommend to add more recent citation such as followings

Response to comment #3:

Thank you for the comment. The reference has been added (Line 86).

Comment #4. Materials and Methods. please add the ethical approval number for animal study

Response to comment #4:

The information has been added (Line 204-205):

“The ethical approval number for animal study â„–3 20.09.2017 (Pacific State Medical University, Russia, Vladivostok).”

Comment #5. Materials and Methods. please describe in more detail about "Synthesis of biocomposite"

Response to comment #5:

The necessary synthetic method explanation have been added in “Materials and Methods” (Line 127-146).

Comment #6. Figure 1 and 2. figure quality(resolution) should be improved. The arrow can be changed to triangle or asterix.

Response to comment #6:

All corrections have been done (Line 228-229 and 248-2249).

Comment #7. Conclusion. Please describe the Conclusion section more briefly.

Response to comment #7:

The “Conclusions” have been corrected (Line 334-340).

Reviewer #2.

This manuscript is a study of the osteoplastic potential of a synthetic CaSiO3/HAp powder biocomposite. The topic is highly relevant and knowledge is needed. However, there are some issues that need to be addressed.

Comment #1. Line 20  - “The authors evaluated the activation possibilities of the osteoinduction process and osteoconduction during the restoration of the volume in the bone tissue in the area of the mandible defect of a laboratory animal” please rephrase this sentence.

Response to comment #1:

Correction has been provided (Line 20-22):

“The authors investigated of the osteoinduction and osteoconduction “in vivo” processes during of the bone tissue regeneration in the mandible defect area of an experimental animal (rabbit)”.

Comment #2. Line 43 – “the lower alveolar ridges” do you mean mandibular ridges?

Response to comment #2:

Should be used considered “the alveolar ridge of lower jaw” (Line 43).

Comment #3. Line 62 – “The solution to this problem”  - what problem do you mean?

Response to comment #3:

Thank you very much for the comment. This sentence is indicated in this context by mistake. The sentence has been deleted.

Comment #4. Line 86 “Synthetic wollastonite is prone to osteoconductivity and bioresorption due to the exchange of bioorganic medium with Ca2+ and SiO32-, because of this it is considered as an artificial graft for bone tissue.” – this sentence is unclear.

Response to comment #4:

The correction of sentenced and necessary references have been provided (Line 94-96):

“Synthetic wollastonite is prone to osteoconductivity and bioresorption due to the exchange of bioorganic medium with Ca2+ and SiO32-. During this biochemical process the bioactive apatite and its derivatives can be produced on the wollastonite surface [32]. Due to this such the synthetic implant is prospective  for the wide applicant as an artificial graft for bone tissue”

Comment #5. Line 88 – “Moreover, wollastonite exerts an expressed angiogenic potential, which is the key position for the osteointegration of the graft” – pleas add a bibliography to support this thesis.

Response to comment #5:

The additional references [doi:10.1016/j.tox.2011.06.016.; https://doi.org/10.1155/2019/1202159; doi: 10.1007/s10561-011-9285-x] have been added in the manuscript [33] [34] [35] (Line 97).

Comment #6. Line 90 – “This occurs because hydroxyapatite is characterized by expressed osteoinductive and osteoconductive properties.” - – pleas add a bibliography to support this thesis.

Response to comment #6:

[https://doi.org/10.1155/2019/1202159; doi: 10.1007/s10561-011-9285-x] have been added in the manuscript [34] [35] (Line 97).

Comment #7. Line 91 – “It stimulates the appearance of VGFR expressing endothelial cells, which is identical to neoangiogenesis in a graft.” – what do you mean by “appearance”?

Response to comment #7:

It means “producing of VGFR…”. The correction has been done (Line 84).

Comment #8.  Line 104 - CaSiO3/Hap -  please change into - CaSiO3/Hap

Response to comment #8:

Correction has been done.

Comment #9. Line 130 – “Powder graft (0.2-0.25 g) with 0.1-0.5 mm fraction” – the sentence is unclear

Response to comment #9:

Correction has been done (Line 162):

“Powder sample with mass 0.2-0.25 g with 0.1-0.5 mm fraction was placed into a formed alveolar socket.”

Comment #10. Line 137 and 139 – Days – please change into days

Response to comment #10:

Correction has been done (Lines 169, 171, 310).

Comment #11. Line 155 - Ct.Wi, μm). – please add “(“

Response to comment #11:

Correction has been done (Line 187).

Comment #12. Lines 153-160 – please use one consisted way of using the brackets

Response to comment #12:

Correction has been done (Line 186-193).

Comment #13. Line 170 – “alveolar well” do you mean alveolar wall?

Response to comment #13:

Absolutely right - Correction has been provided (Line 222).

Comment #14. Lines 172 – “2 statistically significant differences in relation to the control group when Ñ€≥0,5” - what kind of statistical analysis was used and how is it possible that the statistical significance was noted for Ñ€≥0,5? What was the control group? How did you manage to analyze the statistical significance of a group of 4 rabbits?

Response to comment #14:

Morphometric was performed after 10 measurements for each parameter. For statistical analysis we used the Mann-Whitney and Wilcoxon tests. The alveolar holes after extraction of teeth without graft were studied as a control test. For 8 experimental samples (alveolar holes of lower jaws which included graft) after implantation and extraction were analyzed.

If p≥0.5, the differences were considered statistically unreliable.

Correction has been done (Line 194-197 and 223-224).

Comment #15. Fig. 2. A. Magnification is missing. What do the arrows point at?

Response to comment #15:

All corrections have been done (Line 228-236).

Comment #16. Fig.2. C. It is difficult to understand which arrows point at which feature.

Response to comment #16:

All corrections have been done (Line 228-236).

Comment #17. Fig.2.D. It is difficult to understand which arrows point at which feature.

Response to comment #17:

All corrections have been done (Line 228-236).

Comment #18. Line 210 – “Morphologically, the area of the exposure of the graft was characterized by the formation of lamellar compact bone that formed the cortical part of the socket (Figure 3A)” – in Fig.3 legend, Fig 3A is reported as the intact side and 3B as a side with the graft.

Response to comment #18:

All corrections have been done (Line 248-257).

Comment #19. Fig.3. – “C” is missing on the third figure.

Response to comment #19:

All corrections have been done (Line 248-257).

Comment #20. Line 217 – it is challenging to see the increased thickness of the periodontal ligament on Fig.3B same as  the number of osteons on Fig. 3C – Line 225. It seems that Figure 3 is not the figure that is described in the text

Response to comment #20:

All corrections have been done (Line 248-257).

Comment #21. Line 234 – there is no Fig. 3D. It seems that Figure 3 is not the figure that is described in the text.

Response to comment #21:

All corrections have been done (Line 248-257).

Comment #22. Line 241 and 249 – The figure that is mentioned in the text seems to be missing.

Response to comment #22:

All corrections have been done (Line 248-257).

Comment #23. Lines 252 and 253 – the “±” is converted in the pdf into some other symbol

Response to comment #23:

Correction has been done (Line 295-296).

Comment #24. Line 256 – “A vast review showed that the presence of silica stimulated the development of bone matrix, collagen type I, differentiation of osteoblasts, and contributed to the settlement of hydroxyapatite.” - pleas add a bibliography to support this thesis.

Response to comment #24:

The necessary information has been added in the text (Line 301-303):

“The mechanism, characteristics of the interaction of structural components and the phase of bone tissue neoplasm using of silicates are given in the review [doi: 10.1155/2013/141427].

Comment #25. It would be beneficial to change “implant” in the text into “graft”

Response to comment #25:

Correction has been done.

Comment #26. Line 282 – “absorption” should it be “resorption”?

Response to comment #26:

Absolutely right - Correction has been provided (Line 326).

Comment #27. Line 296 – “Thus, it was shown that the studied CaSiO3/HAp biocomposite could be recommended for the application as an osteoplastic material for the rehabilitation of bone defects,  for example, in dentistry, for the prevention of thinning of the periodontium and cortical plate of the compact bone, as well as a decrease in the volume of the bone tissue in the area of the defect, which,  in turn, leads to the deformation of dental arches and alteration of occlusion.” It would be beneficial to divide this sentence into 2 or 3 shorter sentences.

Response to comment #27:

Thank you for the comment. The sentence has been divided (Line 334-340).

Reviewer 3 Report

The manuscript evaluated the influence of synthetic CaSiO3/HAp powder biocomposite on the process of regeneration in the osseous tissue in the alveolar ridges in terms of the morphological characteristics of osteoplastic potential.

The study is of sound design and of clear practical and clinical interest, but some improvements are needed. I suggest accepting this article with minor revision.

I would give some  comments and suggestions, which will help to improve the quality of this manuscript.

1. Introduction. please describe the difference and novelty compared to the previous studies(other authors)

2. Introduction. please clarify and emphasize the objective of this study (last two sentences of the introduction section)

3. Introduction. page 2 74-76. I recommend to add more recent citation such as followings

Kim, J.-W.; Yang, B.-E.; Hong, S.-J.; Choi, H.-G.; Byeon, S.-J.; Lim, H.-K.; Chung, S.-M.; Lee, J.-H.; Byun, S.-H. Bone Regeneration Capability of 3D Printed Ceramic Scaffolds. Int. J. Mol. Sci. 2020, 21, 4837.

4. Materials and Methods. please add the ethical approval number for animal study

5. Materials and Methods. please describe in more detail about "Synthesis of biocomposite"

6. Figure 1 and 2. figure quality(resolution) should be improved. The arrow can be changed to triangle or asterix.

7. Conclusion. Please describe the Conclusion section more briefly.

Good work
Thank you 

Author Response

Reviewer #3.

The manuscript evaluated the influence of synthetic CaSiO3/HAp powder biocomposite on the process of regeneration in the osseous tissue in the alveolar ridges in terms of the morphological characteristics of osteoplastic potential.

The study is of sound design and of clear practical and clinical interest, but some improvements are needed. I suggest accepting this article with minor revision.

I would give some  comments and suggestions, which will help to improve the quality of this manuscript.

Comment #1. Introduction. please describe the difference and novelty compared to the previous studies(other authors)

Response to comment #1:

The necessary information has been added in the manuscript “Introduction” (Line 79-106).

Comment #2. Introduction. please clarify and emphasize the objective of this study (last two sentences of the introduction section)

Response to comment #2:

The necessary correction has been added in the “Introduction” (Line 118-123).

Comment #3. Introduction. page 2 74-76. I recommend to add more recent citation such as followings

Response to comment #3:

Thank you for the comment. The reference has been added (Line 86).

Comment #4. Materials and Methods. please add the ethical approval number for animal study

Response to comment #4:

The information has been added (Line 204-205):

“The ethical approval number for animal study â„–3 20.09.2017 (Pacific State Medical University, Russia, Vladivostok).”

Comment #5. Materials and Methods. please describe in more detail about "Synthesis of biocomposite"

Response to comment #5:

The necessary synthetic method explanation have been added in “Materials and Methods” (Line 127-146).

Comment #6. Figure 1 and 2. figure quality(resolution) should be improved. The arrow can be changed to triangle or asterix.

Response to comment #6:

All corrections have been done (Line 228-229 and 248-2249).

Comment #7. Conclusion. Please describe the Conclusion section more briefly.

Response to comment #7:

The “Conclusions” have been corrected (Line 334-340).
